# Occlusion resistant learning of intuitive physics from videos

## Abstract

To reach human performance on complex tasks, a key ability for artificial systems is to understand physical interactions between objects, and predict future outcomes of a situation. This ability, often referred to as *intuitive physics*, has recently received attention and several methods were proposed to learn these physical rules from video sequences. Yet, most these methods are restricted to the case where no occlusions occur, narrowing the potential areas of application. The main contribution of this paper is a method combining a predictor of object dynamics and a neural renderer efficiently predicting future trajectories and explicitly modelling partial and full occlusions among objects. We present a training procedure enabling learning intuitive physics directly from the input videos containing segmentation masks of objects and their depth. Our results show that our model learns object dynamics despite significant inter-object occlusions, and realistically predicts segmentation masks up to 30 frames in the future. We study model performance for increasing levels of occlusions, and compare results to previous work on the tasks of future prediction and object following. We also show results on predicting motion of objects in real videos and demonstrate significant improvements over state-of-the-art on the object permanence task in the intuitive physics benchmark of Riochet et al. (2018).

## 1 Introduction

Learning intuitive physics has recently raised significant interest in the machine learning literature. To reach human performance on complex visual tasks, artificial systems need to understand the world in terms of macroscopic objects, movements, interactions, etc. Infant development experiments show that young infants quickly acquire an intuitive grasp of how objects interact in the world, and that they use these intuitions for prediction and action planning (Carey, 2009; Baillargeon & Carey, 2012). This includes the notions of gravity (Carey, 2009), continuity of trajectories (Spelke et al., 1995), collisions (Saxe & Carey, 2006), etc. Object permanence, the fact that an object continues to exist when it is occluded, (Kellman & Spelke, 1983), is one of the first concepts developed by infants.

From a modeling point of view, the key scientific question is how to develop general-purpose methods that can make physical predictions in noisy environments, where many variables of the system are unknown. A model that could mimic even some of infant's ability to predict the dynamics of objects and their interactions would be a significant advancement in model-based action planning for robotics (Agrawal et al., 2016; Finn & Levine, 2017). Importantly, to be applied to real-world problems, such a model needs to predict object motion in 3D and handle frequent inter-object occlusions. Yet, to our knowledge, most current works on learning intuitive physics get around this challenge by either i) working in 2D spaces with no occlusions (Battaglia et al., 2016; Chang et al., 2016; Fragkiadaki et al., 2015) or ii) learning end-to-end models without decomposing the scene into objects (Agrawal et al., 2016; Lerer et al., 2016; Finn et al., 2016). The former methods have demonstrated that learning models of intuitive physics is possible but assume ground truth positions of objects are available at both training and test time. The latter methods can operate directly on pixel inputs without knowing ground truth positions of objects but are typically limited to a small number of objects and generalize poorly to new setups (e.g. a new number of objects in the scene, see (Lerer et al., 2016)). A third class of methods has recently

emerged (Janner et al., 2018) that first decomposes the input image of the 3D scene into layers corresponding to masks of individual objects and learns scene dynamics given such object-centric decomposition. Note that here the object dynamics is learnt from pixel masks of individual objects, rather than their ground truth positions. This is difficult for 3D scenes due to frequent inter-object occlusions that present two major challenges. First, estimating accurate position and velocity of objects is challenging due to partial occlusions by other objects. Second, objects can be fully occluded by other objects for a significant number of frames. This work falls into the third class of compositional methods, but develops an occlusion resistant model for learning intuitive physics that addresses both of these challenges due to inter-object occlusions.

In detail we propose a compositional model that from object instance masks and depth fields in two consecutive frames, $(M_{t,t+1}, D_{t,t+1})$, estimates the center, velocity and size of objects. This predicted state $\hat{s}_t$ is then used as input of a *Recurrent Interaction Network*, which predicts a sequence of futures states $\hat{s}_{t+2}, ..., \hat{s}_{t+L}$. This sequence of states is given to the *Compositional Rendering Network* which produces segmentation masks $\hat{M}_{t+2}, ..., \hat{M}_{t+L}$ and depth estimates $\hat{D}_{t+2}, ..., \hat{D}_{t+L}$ in future frames. The key innovation of the proposed model is dealing with partial and complete occlusions in the scene. To deal with partial occlusions, the obtained sequence of masks+depths is compared to the ground truth, and gradients are backpropagated through the pre-trained *Compositional Rendering Network* to refine state predictions. This allows us to refine positions of partially occluded objects where simply taking the centroid of the observed portion of the mask results in an incorrect estimate of the object position. With this refinement object positions are corrected taking into account the unobserved (occluded) portion of the object. The refined state estimates $\bar{s}_{t+1}, ..., \bar{s}_{t+L}$ are used at training time for learning parameters of the *Recurrent Interaction Network* and at test time to improve accuracy of object position prediction when following partially occluded objects. To deal with full occlusions, when the object is not visible in multiple frames, we use the learnt model of object dynamics (*Recurrent Interaction Network*) to predict the position of the object multiple frames ahead and thus recovering the object position after the occlusion.

Using the proposed approach, we show that it is possible to learn object dynamics in 3D environments with severe inter-object occlusions and predict segmentation masks up to 30 frames in the future despite occlusion other objects thus mimicking object permanence.

## 2 RELATED WORK

**Forward modelling in videos.** Forward modelling in video has been studied for action planning (Ebert et al., 2018; Finn et al., 2016) and as a scheme for unsupervised learning of visual features (Lan et al., 2014; Mathieu et al., 2015). In that setup, a model is given a sequence of frames and has to generate frames in future time steps. To succeed in this task, such models need to predict object movements, suggesting that they need to learn physical regularities from video. However, models for end-to-end future frame prediction tend to perform poorly on long-term prediction tasks (say more 5-8 frames (Lan et al., 2014; Mathieu et al., 2015; Finn et al., 2016)), failing to preserve object properties and generating blurry outputs. This suggests that models for intuitive physics may require a more structured representation of objects and their interactions.

**Learning dynamics of objects.** Longer term predictions can be more successful when done on the level of trajectories of individual objects. For example, in (Wu et al., 2017b), the authors propose "scene de-rendering", a system that builds an object-based, structured representation from a static (synthetic) image. The recovered state can be further used for physical reasoning and future prediction using a physics engine on both synthetic and real data (Battaglia et al., 2013; Wu et al., 2017a). Future prediction from static image is often multi-modal (e.g. car can move forward or backward) and hence models able to predict multiple possible future predictions, e.g. based on variational auto-encoders (Xue et al., 2016), are needed.

Others have developed structured models that factor object motion and object rendering into two learnable modules. Examples include (Watters et al., 2017; Marco Fraccaro, 2017; Ehrhardt et al., 2017b;a) that combine object-centric dynamic models and visual encoders.

Such models parse each frame into a set of object state representations, which are used as input of a "dynamic" model, predicting object motion. However, (Marco Fraccaro, 2017) restrict drastically the complexity of the visual input by working on binary 32x32 frames, and (Ehrhardt et al., 2017b;a; Watters et al., 2017) still need ground truth position of objects to train their models. None of these work explicitly models inter-object occlusions, which is the focus of our method. In our work, we build on learnable models of object dynamics (Battaglia et al., 2016) and (Chang et al., 2016), which have the key property that they are compositional and hence can model a variable number of objects, but extend them to learn from visual input rather than ground truth object state vectors. Our work is related to (Janner et al., 2018), done independently and concurrently with our work, who develop an object-oriented model of dynamics coupled with a differentiable object renderer to predict a single image with segmentation masks of objects in a future time, given a single still image as input. In contrast, our model predicts frame-by-frame object motion in scenes with partial and full object occlusion. This is possible because (i) our model of dynamics is recursive, predicting a whole sequence of object movements (instead of one single image in future (Janner et al., 2018)) that allows the model to be applied recursively to follow an object through complete occlusion by other objects; (ii) we design a refinement procedure that allows to refine the estimated positions of objects in case of partial occlusions. In addition, in contrast to (Janner et al., 2018) our model predicts velocity of objects and depth of the scene (also taking as input a pair of frames and the depth field).

Others have proposed unsupervised methods to discover objects and their interactions in 2d videos (van Steenkiste et al., 2018). It is also possible to construct Hierarchical Relation Networks (Mrowca et al., 2018), representing objects as graphs and predicting interactions between pairs of objects. However, this task is still challenging and requires full supervision in the form of ground truth position and velocity of objects.

**Learning physical properties from visual inputs.** Related are also methods for learning physical properties of objects. Learning of physical properties, such as mass, volume or coefficients of friction and restitution, has been considered in (Wu et al., 2016). Others have looked at predicting the stability and/or the dynamics of towers of blocks (Lerer et al., 2016; Zhang et al., 2016; Li et al., 2016a;b; Mirza et al., 2017; Groth et al., 2018). Our work is complementary. We don't consider prediction of physical properties but focus on learning models of object dynamics handling inter-object occlusions at both training and test time. (Greff et al., 2019)

**Contributions.** We describe a model that learns complex dynamics of objects in a 3D environment, where inter-object occlusions occur frequently. Our model combines an abstract representation of the scene (position, velocity and depth of objects), with a compositional neural renderer predicting the resulting object masks with depth and explicitly modelling occlusions between objects. This procedure allows us to train the model even when some objects are partially or totally occluded. Unlike (Watters et al., 2017), our model is fully compositional and handles variable number of objects in the scene. Moreover, it does not require as input annotated inter-frame correspondences during training.

## 3 Occlusion resistant modeling for intuitive physics

This section describes our model for occlusion resistant learning of intuitive physics. We first describe the learning set-up considered in this work. We then describe in detail the two main components of our model. In section 3.2 we outline the compositional renderer with occlusion reasoning that predicts object masks given a scene state representation, and in section 3.3 we detail the recurrent interaction network that predicts the scene state evolution over time. Finally, in section 3.4 we outline the training procedure.

### 3.1 Set-up overview

As illustrated in Figure 1 (and Algorithm in the Supplementary Material), during learning our method observes a sequence of object instance masks and depth fields $M_{t,..,t+L}, D_{t,..,t+L}$. The mask for each frame is composed of a set of channels where each channel represents pixels corresponding to an individual object, along with their color and shape (boxes or balls of different sizes). The model does not require the knowledge of correspondence between

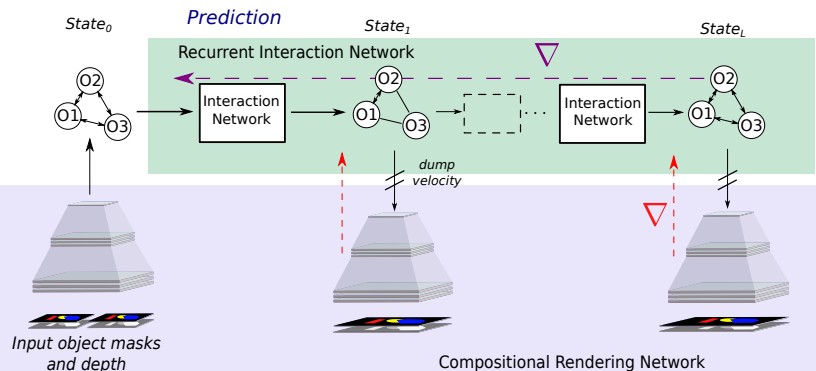

Figure 1: **Training of our occlusion resistant intuitive physics model.** Our model takes as input a video sequence containing segmentation masks of objects in individual frames (bottom). After having estimated the State of the scene consisting of object positions, velocities and their intrinsic properties (shape, color, size) at time 0, the *Recurrent Interaction Network* predicts the state of all objects in the following $L$ frames. The *Compositional Rendering Network* predicts corresponding masks taking into account inter-object occlusions, and errors are backprogated (red arrow) through its frozen weights to refine the object position estimates. These refined states are then used to train (magenta arrow) the paramaters of the *Recurrent Interaction Network*.

objects over time, which might be difficult to obtain in practice. Our model is composed of two networks described below: a pre-trained occlusion sensitive *Compositional Rendering Network (Renderer)* which renders masks and depth fields given a set of object positions (also called states), and a trainable *Recurrent Interaction Network (RecIntNet)* which predicts positions of objects in future frames.

### 3.2 Occlusion modeling: the Compositional Rendering Network

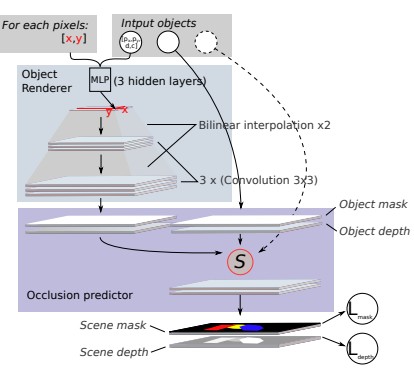

Figure 2: **Compositional Rendering Network.**

We introduce a differentiable *Compositional Rendering Network* (or *Renderer*) that predicts a segmentation mask in the image given a list of $N$ objects specified by their $x$ and $y$ position in the image, depth and possibly additional properties such as object type (e.g. sphere, square, ...) or size. Importantly, our neural rendering model has the ability to take a variable number of objects as input and is invariant to the order of objects in the input list.

**Renderer** (Figure 2) contains two modules. The first module, called the *object rendering network* reconstructs a segmentation mask and a depth map for each object. The second module, called the *occlusion predictor* composes the $N$ predicted object masks into the final scene mask, generating the appropriate pattern of inter-object occlusions obtained from the predicted depth maps of the individual objects.

**The Object rendering network** takes as input a vector of $l$ values corresponding to the position coordinates $(x^k, y^k, d^k)$ of object $k$ in a frame together with additional dimensions for intrinsic object properties (shape, color and size) ($\boldsymbol{c}$). The network predicts object's binary mask, $M^k$ as well as the depth map $D^k$. The input vector $(x^k, y^k, d^k, \boldsymbol{c}^k) \in \mathbb{R}^l$ is first copied into a $(l+2) \times 16 \times 16$ tensor, where each $16 \times 16$ cell position contains an identical copy of the input vector together with $x$ and $y$ coordinates of the cell. Adding the $x$ and $y$ coordinates may seem redundant, but this kind of *position field* enables a very local computation of the shape of the object and avoids a large number of network parameters (similar architectures were recently also studied in (**?**)).

The input tensor is processed with $1 \times 1$ convolution filters. The resulting 16-channel feature map is further processed by three blocks of convolutions. Each block contains three convolutions with filters of size $1 \times 1$, $3 \times 3$ and $1 \times 1$ respectively, and 4, 4 and 16 feature maps, respectively. We use `ReLU` pre-activation before each convolution, and up-sample (scale of 2 and bilinear interpolation) feature maps between blocks. The last convolution outputs $N + 1$ feature maps of size $128 \times 128$, the first feature map encoding depth and the $N$ last feature maps encoding mask predictions for the individual objects. The object rendering network is applied to all objects present, resulting in a set of masks and depth maps denoted as $\{(\hat{M}^k, \hat{D}^k), k = 1..N\}$.

**The Occlusion predictor** takes as input the masks and depth maps for $N$ objects and aggregates them to construct the final occlusion-consistent mask and depth map. To do so it computes, for each pixel $i, j \leq 128$ and object $k$ the following weight:

$$c_{i,j}^k = \frac{e^{\lambda \hat{D}_{i,j}^k}}{\sum_{q=1}^{N} e^{\lambda \hat{D}_{i,j}^q}}, k = 1..N, \tag{1}$$

where $\lambda$ is a parameter learned by the model. The final masks and depth maps are computed as a weighted combination of masks $\hat{M}_{i,j}^k$ and depth maps $\hat{D}_{i,j}^k$ for individual objects $k$: $\hat{M}_{i,j} = \sum_{k=1}^{N} c_{i,j}^k \hat{M}_{i,j}^k$, $\hat{D}_{i,j} = \sum_{k=1}^{N} c_{i,j}^k \hat{D}_{i,j}^k$, where $i, j$ are output pixel coordinates $\forall i, j \leq 128$ and $c_{i,j}^k$ the weights given by (1). The intuition is that the occlusion renderer constructs the final output $(\hat{M}, \hat{D})$ by selecting, for every pixel, the mask with minimal depth (corresponding to the object occluding all other objects). For negative values of $\lambda$ equation (1) is as a softmin, that selects for every pixel the object with minimal predicted depth. Because $\lambda$ is a trainable parameter, gradient descent forces it to take large negative values, ensuring good occlusion predictions. Also note that this model does not require to be supervised by the depth field to predict occlusions correctly. In this case, the object rendering network still predicts a feature map $\hat{D}$ that is not equal to the depth anymore but is rather an abstract quantity that preserves the relative order of objects in the view. This allows *Renderer* to predict occlusions when the target masks are RGB only. However, it still needs depth information about in the input (either true depth or relative ordering).

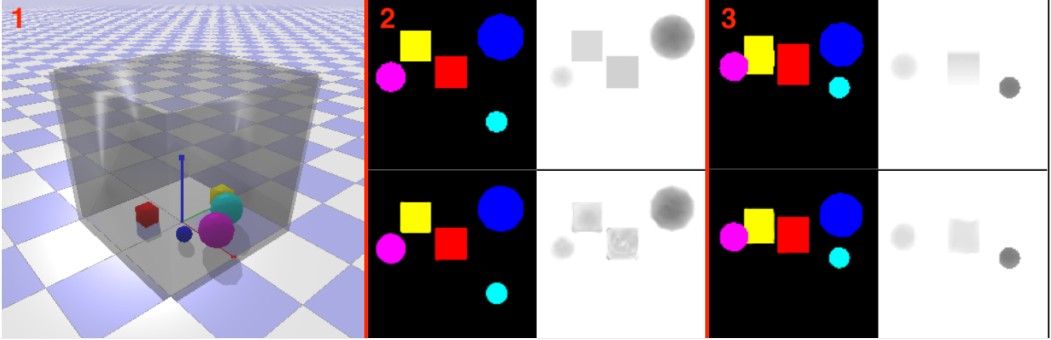

Figure 3: 1: An overview of the scene. 2&3: Sample video frames (instance mask + depth field) from our datasets (top) together with predictions obtained by our model (bottom). Taken from the top-view experiment (2) and tilted 25° experiments (3). **Please see additional video results in the anonymous google drive** `https://drive.google.com/drive/folders/111XK6GZnmHjd_7US6LGkxg6cAhJ2WDxQ?usp=sharing.`

### 3.3 Dynamics prediction: the Recurrent Interaction Network (*RecIntNet*)

To model object dynamics, we build on the Interaction Network (Battaglia et al., 2016), which predicts dynamics of a variable number of objects by modelling their pairwise interactions. Here we describe three extensions of the vanilla Interaction Network model. First, we extend the Interaction Network to model 2.5D scenes where position and velocity have a depth component. Second, we extend the Interaction Network to train from the whole sequence of future states and call this new model Recurrent Interaction Network. Third, we introduce

variance in the position predictions, to stabilise the learning phase, and avoid penalizing too much very encertain predictions. The three extensions are described below.

**Modelling compositional object dynamics in 2.5D scenes.** As shown in (Battaglia et al., 2016), Interaction Networks can be used to predict object motion both in 3D or in 2D space. Given a list of objects represented by their positions, velocities and size in the Cartesian plane, an Interaction Network models interactions between all pairs of objects, aggregates them over the image and predicts the resulting motion for each object. Here, we model object interactions in 2.5D space, since we have no access to the object position and velocity in the Cartesian space. Instead we have locations and velocities in the image plane plus depth (the distance between the objects and the camera).

**Training from a sequence of future frames.** The vanilla Interaction Network (Battaglia et al., 2016) is trained to predict position and velocity of each object in one step into the future. Here, we learn from multiple future frames. In detail, we "rollout" the Interaction Network to predict a whole sequence of future states as if a standard Interaction Network was applied in recurrent manner. We found that faster training can be achieved by directly predicting changes in the velocity, hence:

$$[p_1, v_1, c] = [p_0 + \delta t v_0 + \frac{\delta t^2}{2}\mathbf{d_v}, v_0 + \mathbf{d_v}, c], \tag{2}$$

where $p_1$ and $v_1$ are position and velocity of the object at time $t_1$, $p_0$ and $v_0$ are position and velocity at time $t_0$, and $\delta t = t_1 - t_0$ is the time step. Position and velocity in pixel space ($p = [p_x, p_y, d]$ where $p_x, p_y$ are the position of the object in the frame), $d$ is depth and $v$ is the velocity in that space. Hence $\mathbf{d_v}$ can be seen as the *acceleration*, and $(v_0 + \mathbf{d_v}), (p_0 + \delta t v_0 + \frac{\delta t^2}{2}\mathbf{d_v})$ as the first and second order Taylor approximations of velocity and position, respectively. Assuming an initial weight distribution close to zero, this gives the model a prior that the object motion is linear.

**Prediction uncertainty.** To account for prediction uncertainty and stabilize learning, we assume that object position follows a multivariate normal distribution, with diagonal covariance matrix. Each term $\sigma_x^2, \sigma_y^2, \sigma_d^2$ of the covariance matrix represents the uncertainty in prediction, along x-axis, y-axis and depth. Such uncertainty is also given as input to the model, to account for uncertainty either in object detection (first prediction step) or in the recurrent object state prediction. The resulting loss is negative log-likelihood of the target $p_1$ w.r.t. the multivariate normal distribution, which reduces to

$$\mathcal{L}\big((\hat{p_1}, \hat{\tau_1}), p_1\big) = \frac{(\hat{p_1} - p_1)^2}{\exp \hat{\tau_1}} + \hat{\tau_1}, \tag{3}$$

where $\hat{\tau_1} = \log \hat{\sigma_1^2}$ is the estimated level of noise propagated through the Recurrent Interaction Network, where $\sigma_1$ concatenates $\sigma_x^2, \sigma_y^2, \sigma_d^2$, $p_1$ is the ground truth state and $\hat{p_1}$ is the predicted state at time $t + 1$. The intuition is that the squared error term in the numerator is weighted by the estimated level of noise $\hat{\tau_1}$, which acts also as an additional regularizer.

## 3.4 System Training

In this section we give a high level description of the procedure for training our model (details are in the supplementary Section S3). The Compositional rendering Network is pre-trained offline from masks and depths in individual frames. Training of the Recurrent Interaction Network is done in the following three steps. First (*initialization phase*), we select from each training video short clips containing $L$ frames (here, we use L=10). In each frame, we estimate position, depth and size of each object by computing the centroid of each mask, its average depth and diameter (max distance between two mask pixels). To correct errors due to partial occlusions, we perform *occlusion-aware refinement of object positions* using gradient descent through the pre-trained *Renderer* (see supplementary material). The result is a partial state vector (no velocities) for each frame, corrected for partial occlusions. Spatially close objects in two consecutive frames are linked and considered the same objects. In a second step (*prediction phase*), we use the Recurrent Interaction Network to roll out $L - 2$ predictions for the position of these objects in future frames starting from Frame 2. Frame 1 is used to compute the initial velocities of Frame 2. In a third step, (*update phase*), we use the distance between the ground truth positions established in step 1 and the rollout positions to perform the training of the Recurrent Interaction Network.

## 4 EXPERIMENTS

In this section we demonstrate the ability of our model to learn intuitive physics in presence of inter-object occlusions. We evaluate our model on two task: (i) *future prediction*, predicting objects' trajectories up to a horizon of 10 frames and (ii) *object following*, coupling the dynamics network with the neural renderer to follow objects under occlusions, up to a horizon of 30 frames. In *future prediction*, we initialize the network with two frames, which enable the computation of object positions and velocities based on the instance masks as in the training phase. We then run a roll-out for N consecutive frames with the interaction network. We evaluate this rollout by comparing the predicted positions or reconstructed pixels with the ground truth. In *object following*, we alternate between short-term rollout (using the interaction network to predict the next frame) and object position refinement (using the renderer). This allows us to put an index on each object, and follow them through large periods of occlusions. During full occlusion, the position is solely determined by the interaction network, since the object position refinement has a zero gradient. During full or partial occlusion, object position refinement is used to reconstruct a better estimate of the positions and velocities. To test object following, we measure the accuracy of the position estimates across long sequences containing occlusions. We also evaluate the ability to detect the violation of object permanence (objects disappearing or appearing out of nowhere). Our evaluation is mostly based on a synthetic dataset, which we release for this paper. We also study generalization to real scenes, and compare to baseline models on the object permanance subset of the intuitive physics benchmark (Riochet et al., 2018).

### 4.1 EVALUATING FUTURE PREDICTION

We use pybullet[1] physics simulator to generate videos of variable number of balls of different colors and sizes bouncing in a 3D scene (a large box with solid walls) containing a variable number of smaller static 3D boxes. We generate five datasets, where we vary the camera tilt and the presence of occluders. In the first dataset ("Top view") we record videos with a top camera view (or 90°), where the borders of the frame coincide with the walls of the box. In the second dataset ("Top view+occ"), we add a large moving object occluding 25% of the scene. Finally, we decrease the camera viewing angle to 45°, 25° and 15° degrees, which results in an increasing amount of inter-object object occlusions due to perspective projection of the 3D scene onto a 2D image plane. We computed the proportion of time each object is occluded or partially occluded and found 3.1% in the top-view videos, 31.1% in the top view occluded videos, and 5.9%, 11.7%, 13.4% in the 45°, 25°, 15° tilted videos, respectively. Additional details of the datasets are given in the supplementary material.

**Inter-object occlusion investigation.** In this section we consider prediction horizons of 5 and 10 frames, and evaluate the position error as a L2 distance between the predicted and target object positions. L2 distance is computed in the 3D Cartesian scene coordinates, such that results are comparable across different camera tilts. Results are shown in Table 1. We first note that our model trained on mask and depth prediction significantly outperforms the linear baseline, which is computed as an extrapolation of the position of objects based on their initial velocities. Moreover, the results of our method are relatively stable across challenging setups with occlusions by external objects or frequent self-occlusions in tilted views. This demonstrates the potential ability of our method to be trained from real videos where occlusions and other factors usually prevent reliable recovery of object states.

**Ablation Studies.** As an ablation study we replace the Recurrent Interaction Network (*RecIntNet*) in our model with a multi-layer perceptron. This MLP contains four hidden layers of size 180 and is trained the same way as *RecIntNet*, modelling acceleration as described in equation 3.3. To deal with the varying number of objects in the dataset, we pad the inputs with zeros. We observe that *RecIntNet* allows more robust predictions through time.

As a second ablation study, we train the Recurrent Interaction Network without modelling acceleration (3.3). This is similar to the model described in (Janner et al., 2018), where object representation is not decomposed into position / velocity / intrinsic properties, but is

---

[1]https://pypi.org/project/pybullet

|  | Top view | Top view+occ. | 45° tilt | 25° tilt | 15° tilt |
|---|---|---|---|---|---|
| Trained on positions inferred from masks and depth | | | | | |
| Linear baseline | 47.6 / 106.0 | 47.6 /106.0 | 47.6 / 106.0 | 47.6 / 106.0 | 47.6 / 106.0 |
| MLP baseline | 13.1 / 15.7 | 17.3 / 19.2 | 18.1 / 23.8 | 17.6 / 24.6 | 19.4 / 26.2 |
| NoDyn-RIN | 21.2 / 46.2 | 23.7 / 46.7 | 22.5 / 42.8 | 23.1 / 43.3 | 24.9 / 44.4 |
| NoProba-RIN | 6.3 / 11.5 | 12.4 / 14.7 | **8.0** / 15.9 | 8.12 / 16.3 | 11.2 / 19.6 |
| RIN | 6.3 / **9.2** | **11.7** / **13.5** | 8.01 / **14.5** | **8.1** / **15.0** | 11.2 / **18.1** |
| Trained on ground truth positions* | | | | | |
| Proba-RIN | 4.5 / 9.0 | NA | 6.0 / 9.6 | 5.2 / 12.2 | 7.3 / 13.2 |
| (Battaglia et al., 2016)** | 3.6 / 10.1 | NA | 4.5 / 9.9 | 4.5 / 11.0 | 5.3 / 12.3 |

Table 1: Average Euclidean (L2) distance (in pixels) between predicted and ground truth positions, for a prediction horizon of 5 frames / 10 frames. To compute the distance, the pixel-based x-y-d coordinates of objects are projected back in an untilted 200x200x200 reference Cartesian coordinate system. Methods denoted by * have access to the ground truth object positions at all time, even behind occluders. Their performance is hence optimistic and not directly comparable to the other methods trained only from observed object masks and depth fields. **(Battaglia et al., 2016) is trained with more supervision, since target values include ground truth velocities, not available to other methods.

rather a (unstructured) 256-dimensional vector. We observe a significant loss in performance, tending to confirm that modelling position and velocity explicity, and having a constant velocity prior on motion (given by 3.3) improves future predictions.

As a third ablation study, we train a deterministic variant of *RecIntNet*, where only the sequence of states is predicted, without the uncertainty term $\tau$. The loss considered is the mean squared error between the predicted and the observation state. Observed results are slightly worse than our model handling uncertainty (see NoProba-RIN), but close enough to say that this is not a key feature for modelling 5 or 10 frames in the future. In qualitative experiments, however, we observed more robust long-term predictions after introducing the uncertainty term $\tau$ in the model and the loss (equation 3).

For the purpose of comparison, we also evaluate three models trained using ground truth object states. Our Recurrent Interaction Network trained on ground truth object states gives similar results to the model of (Battaglia et al., 2016). As expected, training on ground truth states (effectively ignoring occlusions and other effects) performs better than training from object masks and depth. We also compare with CNN autoencoder (Riochet et al., 2018), showing our models gives better forward mask and depth predictions than CNN auto-encoders trained end-to-end. Full results are given in the supplementary material.

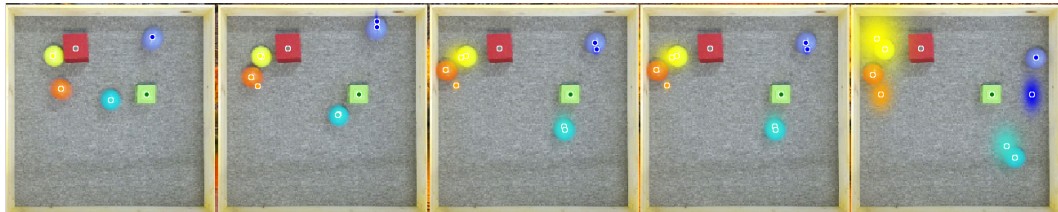

Figure 4: The figure shows example predictions in five frames in a real scene containing 10 frames. The small colored dots show the predicted positions of objects together with the estimated uncertainty shown by the colored "cloud". The same colored dot is also shown in the (ground truth) center of each object. The prediction is correct when the two dots coincide. (see additional videos).

**Generalization to real scenes.** We construct a dataset of 22 real videos, containing a variable number of colored balls and blocks in motion. Videos are recorded with a Microsoft kinect2 device, including RGB and depth frames. The setup is similar to the one generated with pybullet, recorded with a top camera view and containing 4 balls and a variable number of blocs (from 0 to 3). Here again, the borders of the frame coincide with the walls of the

| Model | Linear baseline | MLP | Proba-RIN |
|---|---|---|---|
| L2 distance to target | 28/71 | 19/43 | **12/22** |

Table 2: Average Euclidean (L2) distance (in pixels in a 200 by 200 image) between predicted and ground truth positions, for a prediction horizon of 5 frames / 10 frames.

| Synthetic videos | 5 fr. | 10 fr. | 30 fr. |
|---|---|---|---|
| Ours, top view | 100 | 100 | 100 |
| Ours, 45° tilt | 99.3 | 96.2 | 96.2 |
| Ours, 25° tilt | 99.3 | 90.1 | 90.1 |
| Linear motion baseline | 81.1 | 67.8 | 59.7 |

Table 3: Percentage of predictions within a 20-pixel neighborhood around the target as a function of rollout length measured by the number of frames. 20 pixels corresponds to the size of the smallest objects in the dataset.

box. Taking as input object segmentation of the first two frames, we use our model to predict object trajectories through the whole video (see Figure 4). We use the model trained on top-view pybullet videos, without fine-tuning weights. We measure the error between predictions and ground truth positions along the rollout. Results are shown in Table 2 and clearly demonstrate that out approach outperforms the linear and MLP baselines.

### 4.2 Evaluating object following

**Evaluation on long roll-outs.** We ran at test time longer roll-outs (up to 30 frames), iteratively corrected by our occlusion-aware refinement procedure. This can be viewed as a form of tracking evaluation. Table 3 shows the percentage of object predictions that diverge by more than an object diameter (20 pixels) using this method. The performance is very good, even for tilted views. Supplementary Figure 1 shows these numbers as a function of the pixel threshold. Roll-outs (without refinement) are provided in the following anonymous google drive (link), showing qualitatively convincing trajectories and bouncing behaviors, even for tilted views.

**Evaluation on IntPhys benchmark.** Riochet et al. (2018) propose a benchmark to evaluate intuitive physics models. Drawing inspiration from infant development studies, the benchmark consists of classifying whether a particular video is physically possible or impossible. We focus on the task `O1 / Occluded / Dynamic_1`, evaluating the notion of object permanence in presence of occlusions. From the provided train set [2] containing only possible videos, we train our model to predict a sequence of 8 frames from an input pair of frames. The evaluation subset `O1 / Occluded / Dynamic_1` contains on 720 videos, forming 180 quadruplet of (2 possible / 2 impossible) videos. Starting from the first visible position of an object, we predict its trajectory until the end of the video, refining prediction at every time step. For each video, the predicted masks are compared with the observed masks, resulting in a sequence of reconstruction errors. We derive an implausibility score for a video as the maximum error through the whole sequence. For each quadruplet of (2 possible / 2 impossible) videos, we classify the two videos that have the highest implausibility score as impossible, the two other as possible. Table 4 reports error rates, in comparison with baselines from Riochet et al. (2018). We can see a clear improvement of our method, confirming it can follow objects through long occlusions.

## 5 Discussion

Learning the physics of simple macroscopic object dynamics and interactions is a relatively easy task when ground truth coordinates are provided to the system, and techniques like Interaction Networks trained with a future frame prediction loss are quite successful (Battaglia et al., 2016). Of course a major drawback of this kind of system is that it is basically restricted

---

[2] www.intphys.com

| | Ours | | | | (Riochet et al., 2018) | | | |
| --- | --- | --- | --- | --- | --- | --- | --- | --- |
| | 1 obj. | 2 obj. | 3 obj. | Total | 1 obj. | 2 obj. | 3 obj. | Total |
| Dyn 1 | **0.37** | **0.37** | **0.39** | **0.38** | 0.39 | 0.56 | 0.54 | 0.50 |

Table 4: Detailed results of relative classification error of our model compared to the (Riochet et al., 2018) baseline on the IntPhys object permanence benchmark, block O1 / Occluded / Dynamic 1. Lower is better.

to learning physics from 3D simulators. In real life, the ground truth coordinates of each object are unknown, only projected 2D views are available. Interestingly, we found that projective geometry is not, in and of itself, a difficulty. Indeed, when an Interaction Network is fed, not with 3D Cartesian object coordinates, but with a 2.5D projective referential such as the xy position of objects in a retina (plus depth), the accuracy of the prediction remains unchanged compared with the Cartesian ground truth. As RGBD videos are relatively easy to collect in large quantities, it would not be difficult to train systems with such inputs. But real world videos raise two other major difficulties: (i) images are not easily segmentable into objects, and (ii) objects do not remain always visible and tend to be occluded by other objects. This makes the ground truth coordinates of objects only partially observable.

Here, we provided a first step towards more realistic physics learning by addressing the occlusion problem. We introduce a physics learning system composed of an Interaction Network followed by a trainable Renderer. The Interaction Network has been made recurrent, such that ground truth positions and velocities have only to be fed at the first frame. The renderer can be qualified as 2.5D, in that it takes as input the positions and velocities of objects (in retina pixel xy-d coordinates) and computes the resulting instance masks (plus depth). The 2.5D renderer is itself relatively lightweight (only 1233 parameters). It is based on a rather simple convolutional architecture, uses position fields, and can be trained with few examples to render objects of arbitrary shapes with controlable 2.5D positions respecting occlusions. The outcome can be seen on the rendering of tilted views as shown in the videos provided in the anonymous google drive (link).

What we showed is that instead of training the interaction network to predict ground truth positions, we can directly train it through estimates obtained from mask+depth, corrected through the renderer. The resulting system, of course, produces less accurate predictions that when trained with real positions, but is still better than either linear baselines, or CNN mask prediction networks by (Riochet et al., 2018; Lerer et al., 2016). Interestingly, the reconstruction loss is still effective even in the presence of external occluders, or when objects occlude each other because of a tilted view during training. This can be explained by the fact that when an object is occluded, the gradient of the reconstruction loss will be zero (because no matter where the object is predicted to be, so long as it is predicted to be behind another object, it is not visible, hence contributes to no loss). This amounts to simply reducing the size of the training set, so it only slightly degrades the final performance. Importantly, this cancellation of the losses occurs without explicitly telling the system which objects are occluded and which are not. This is implicitly learnt by the system through the rendering network. Applying this method to the intuitive physics benchmark presented in (Riochet et al., 2018), we show it outperforms baselines on modelling object dynamics under occlusions.

Further work needs to be done to fully train this system end-to-end, in particular, by learning the renderer and the interaction network jointly. Another avenues relate to the first problem raised above, i.e. the segmentation problem. Object segmentation based on raw pixels has been addressed in previous work, but yields errors (over- or under-segmentations) on more realistic datasets, which could have dramatic effect on the interaction network, which crucially depends on reliable object identification. Such issues need to be addressed before end-to-end physics prediction systems can be trained and used with real videos, and approximate the ability of infants to predict the interactions of objects from live or video scenes.

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

# Occlusion resistant learning of intuitive physics from videos
# Supplementary material

**Anonymous authors**

This supplementary material: (i) describes the provided supplementary videos (section ), (ii) gives details of the datasets used in the experiments (section ), (iii) provides additional training details (section ) and (iv) gives more detailed test-time roll-out results (section ).

## S1. Description of supplementary videos

The videos are in the anonymous google drive: `https://drive.google.com/drive/folders/111XK6GZnmHjd_7US6LGkxg6cAhJ2WDxQ?usp=sharing` in the videos/ subdirectory. See also the README slideshow.

- **scene_overview.mp4** shows raw videos of the entire environment.

- **tracking_occlusions_*.mp4** show examples of position prediction through complete occlusions, using our occlusion-aware object position refinement. This shows that our model can keep track of the object identity through complete occlusions, mimicking "object permanence".

- **one_class*.mp4** show different examples of our model following motion of multiple objects in the scene. All balls have the same color which makes them difficult to follow in case of mutual interactions. Videos come from tilted 25° experiments, which are the most challenging because they include inter-object occlusions. Dots represent the predicted position of each object, the color being its identity. Our model shows very good predictions with small colored markers (dots) well centered in the middle of each object, with marker color remaining constant for each object preserving the object identity during occlusions and collisions. **one_class_raw*.mp4** show rendered original views of the same dynamic scenes but imaged from a different viewpoint for better understanding.

- **rollout_0.mp4**, **rollout_1.mp4** show three different rollouts without position refinement. From left to right: ground truth trajectories, our model trained of state, our model trained on masks, our model trained on masks with occlusions during training. Rollout length is 20 frames.

- **rollout_tilt*_model.mp4** and **rollout_tilt*_groundtruth.mp4** show the same dynamic scene but observed with various camera tilts (e.g. **tilt45_model.mp4** show a video for a camera tilt of 45 degrees). ***_model.mp4** are rollouts of our Recurrent Interaction Network (*RecIntNet*) computed without the occlusion-aware position refinement based on the observed masks (pure forward prediction of the dynamics model). ***_groundtruth.mp4** are the corresponding ground-truth trajectories, rendered with the *Compositional Rendering Network*.

- **intphys_*.mp4** show object following in the IntPhys training set.

- **rollout_pybullet_*.mp4** show free rollout (no refinement) on synthetic dataset.

- **rollout_real_*.mp4** show generalization to real scenes.

## S2. Datasets

To validate our model, we use pybullet[1] physics simulator to generate videos of variable number of balls of different colors and sizes bouncing in a 3D scene (a large box with solid walls) containing a variable number of smaller static 3D boxes. We generate five dataset versions, where we vary the camera tilt and the presence of occluders. All experiments are made with datasets of 12,000 videos of 30 frames (with a frame rate of 20 frames per second). For each dataset, we keep 2,000 videos separate to pre-train the renderer, 9,000 videos to train the physics predictor and 1,000 videos for evaluation. Our scene contains a variable number of balls (up to 6) with random initial positions and velocities, bouncing against each other and the walls. Initial positions are sampled from a uniform distribution in the box $[1, 200]^2$, all balls lying on the ground. Initial velocities along $x$ and $y$ axes are sampled in $Unif([-25, 25])$ units per frame, initial velocity along $z$-axis is set to 0. The radius of each ball is sampled uniformly in $[10, 40]$. Scenes also contain a variable number of boxes (up to 2) fixed to the floor, against which balls can collide. Contrary to Battaglia et al. (2016) where authors set a frame rate of 1000 frames per second, we sample 30 frames per second, which is more reasonable when working with masks (because of the computation cost of mask prediction).

**Top-view.** In the first dataset we record videos with a top camera view, where the borders of the frame coincide with the walls of the box. Here, initial motion is orthogonal to the camera, which makes this dataset very similar to the 2D bouncing balls datasets presented in Battaglia et al. (2016) and Watters et al. (2017). However, our dataset is 3D and because of collisions and the fact that the balls have different sizes, balls can jump on top of each other, making occlusions possible, even if not frequent.

**Top-view with Occlusions.** To test the ability of our method to learn object dynamics in environments where occlusions occur frequently, we record the second dataset including frequent occlusions. We add an occluder to the scene, which is an object of irregular shape (an airplane), occluding 25% of the frame and moving in 3D between the balls and the camera. This occluder has a rectilinear motion and goes from the bottom to the top of the frame during the whole video sequence. Sample frames and rendered predictions can be found in the supplementary material.

**Tilted-views.** In three additional datasets we keep the same objects and motions but tilt the camera with angles of $45°$, $65°$ and $75°$ degrees. Increasing the tilt of the camera results in more severe inter-object occlusions (both partial and complete) where the balls pass in front of each other, and in front and behind the static boxes, at different distances to the camera. In addition, the ball trajectories are becoming more complex due to increasing perspective effects. In contrary to the top-view experiment, the motion is not orthogonal to the camera plane anymore, and depth becomes crucial to predict the future motion.

## S3. Training details

This section gives details of the offline Pre-Training of the compositional Rendering Network and detailed outline of the algorithm for training the Recurrent Interaction Network.

**Pre-Training the Compositional Rendering Network.** We train the neural renderer to predict mask and depth $\hat{M}_t, \hat{D}_t$ from a list of objects $[p_x, p_y, d, \boldsymbol{c}]$ where $p_x, p_y$ are x-y coordinates of the object in the frame, $d$ is the distance between the object and the camera and $\boldsymbol{c}$ is a vector for intrinsic object properties containing the size of the object, its class (in our experiments a binary variable for whether the object is a ball, a square or an occluder) and its color as vector in $[0, 1]^3$.

The target mask is a $128 \times 128$ image where each pixel value indicates the index of the corresponding object mask (0 for the background, $i \in 1..N$ for objects). The loss on the

---

[1] https://pypi.org/project/pybullet

mask is negative log-likelihood, which corresponds to the average classification loss on each pixel

$$L_{\text{mask}}(\hat{M}, M) = \sum_{i \leq h, j \leq w} \sum_{n \leq N} \mathbf{1}(M_{i,j} = n) log(\hat{M}_{i,j,n}), \tag{1}$$

where the first sum is over individual pixels indexed by $i$ and $j$, the second sum is over the individual objects indexed by $n$, $\forall \hat{M} \in [0,1]^{h \times w \times N}$ are the predicted (soft-) object masks, and $\forall M \in [\![1, N]\!]^{h \times w}$ is the scene ground truth mask containing all objects. The target depth map is a $128 \times 128$ image with values being normalized to the [-1,1] interval during training. The loss on the depth map prediction is the mean squared error

$$L_{\text{depth}}(\hat{D}, D) = \sum_{i \leq h, j \leq w} (\hat{D}_{i,j} - D_{i,j})^2, \tag{2}$$

where $\forall \hat{D}$ and $D \in \mathbb{R}^{h \times w}$ are the predicted and ground truth depth maps, respectively. The final loss used to train the renderer is the weighted sum of losses on masks and depth maps, $L = 0.7 * L_{mask} + 0.3 * L_{depth}$. We use the `Adam` optimizer with default parameters, and reduce learning rate by a factor 10 each time the loss on the validation set does not decrease during 10 epochs. We pre-train the network on a separate set of 15000 images generated with `pybullet` and containing similar objects as in our videos.

**Training details of the Recurrent Interaction Network.** The detailed outline of training the Recurrent Interaction Network is given in Algorithm 1.

---

**Algorithm 1:** Train Recurrent Interaction Network

**Data:**
    $T, L$: length of the video and prediction span, respectively;
    $M_t$, $t = 1..T$: Instance masks;
    $D_t$, $t = 1..T$: Depth maps;
    `Rend`: Pre-trained Renderer;
    `RIN`: Recurrent Interaction Network (initialized with constant velocity motion);
    `Criterion`: stopping criterion (`RIN` loss on validation);
    `Detection(`$m_t, d_t$`)`: returns centroid, depth and size of instance masks;
    `NLL`, `MSE`: Negative Log-Likelihood and Mean-Squared Error, respectively;
**Result:**
    Trajectory estimates $\bar{p}_{t+1..t+L}$;
    Trained Recurrent Interaction Network $w_{\text{RIN}}$;
**while** *Criterion(RIN)* **do**
    **for** $t \in \{1..T-1\}$ **do** // Initialization of positions and velocities
        // Initial object positions from observed masks and depths
        $\hat{p}_t \leftarrow$ `Detection(`$m_t, d_t$`)`;
        // Occlusion-aware object position refinement using Renderer
        $\bar{p}_t \leftarrow \arg \min_{p \leftarrow \hat{p}_t}$ `NLL(Rend(`$p$`)`, $(m_t, d_t)$`)`;
        // Estimate object velocities from consecutive frames
        $\bar{v}_t \leftarrow \bar{p}_{t+1} - \bar{p}_t$ ;
    **for** $t \in \{1..T-L\}$ **do** // Training Recurrent Interaction Network
        // Predict sequence of states (of all objects) using roll-out
        $\hat{p}_{t+1..t+L} \leftarrow$ `RIN(`$\bar{p}_t, \bar{v}_t$`)`;
        // Occlusion-aware object position refinement
        $\bar{p}_{t+1..t+L} \leftarrow \arg \min_{p \leftarrow \hat{p}_{t+1..t+L}}$ `NLL(Rend(`$p$`)`, $(m_t, d_t)$`)`;
        // Update weights of Recurrent Interaction Network
        $w_{\text{RIN}} \leftarrow \arg \min_w$ `MSE(RIN(`$\bar{p}_t$`)`, $\hat{p}_{t+1..t+L}$`)`;

---

Given an initial state $s_t$, the Recurrent Interaction Network recursively predicts a sequence of future states $\hat{s}_{t+1}, \hat{s}_{t+2}, ..., \hat{s}_{t+L}$, as well as error terms $\hat{\tau}_{t+1}, \hat{\tau}_{t+2}, ..., \hat{\tau}_{t+L}$. This predicted sequence is compared to object positions (ground truth or derived from masks after refinement), and the loss is computed as the sum of negative log likelihood (**??**) along the sequence.

| | Top view | Top view+ occlusion | 45° tilt | 25° tilt | 15° tilt |
|---|---|---|---|---|---|
| CNN autoencoder Riochet et al. (2018) | 0.0147 | 0.0451 | 0.0125 | 0.0124 | 0.0121 |
| RIN, trained on mask+depth | 0.0101 | **0.0342** | 0.0072 | **0.0070** | 0.0069 |
| Proba-RIN, trained on mask+depth | **0.0100** | 0.0351 | **0.0069** | 0.0071 | **0.0065** |

Table S1. Aggregate pixel reconstruction error for mask and depth, for a prediction span of two frames. This error is the loss used for training (described in the supplementary material). It is a weighted combination of mask error (per-pixel classification error) and the depth error (mean squared error).

We use `Adam` optimizer, and divide the learning rate by $L$ to be consistent with the size of the sequence (as the loss is a sum over a sequence of length $L$). The same learning rate decay and stopping procedure is applied. Sequence lengths of 4, 6 and 10 were tested during training, lengths of 10 giving slightly more stable rollouts.

## S4. Occlusion-aware refinement of object positions

Position refinement consists of using the pre-trained *Renderer* to correct estimated positions of all objects in a particular frame. To do so, we give the position estimates as input to the *Renderer* which outputs a corresponding pair of mask and depth field for the frame, $(\hat{M}, \hat{D})$, properly rendering the inter-object occlusions. This prediction is compared to the observed mask and depth field, returning errors that are backpropagated through the frozen weights of the *Renderer*. We perform gradient descent on the input itself to correct object position and size estimates, according to the observations. In our experiments, we set learning rate to 0.01 and compute 200 iterations of gradient descent. Details of the loss are given in the supplementary material.

For object positions estimated from object masks, this refinement allows us to reduce errors due to partial occlusions (moving the predicted center of one object from its visible mask centroid to its real center).

## S4. Future prediction: Comparison with Riochet et al. (2018)

We evaluate the error of the mask and depth prediction, measured by the training error described in detail in . Here, we compare our model to a CNN autoencoder Riochet et al. (2018), which directly predicts future masks from current ones, without explicitly modelling dynamics of the individual objects in the scene. Note this baseline is similar to Lerer et al. (2016). Results are shown in Table S1. As before, the existence of external occluders or the presence of tilt degrades the performance, but even in this case, our model remains much better than the CNN autoencoder of Riochet et al. (2018).

## S5. Detailed roll-out results

In Figure S1, we report the proportion of correctly followed objects for different rollout lengths (5, 10 and 30 frames) as a function of the distance error (pixels). Note that the size of the smallest object is around 20 pixels.

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

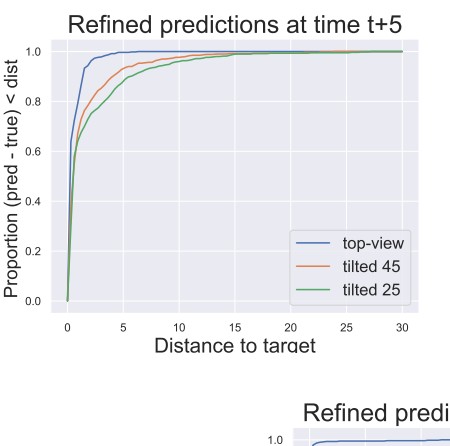

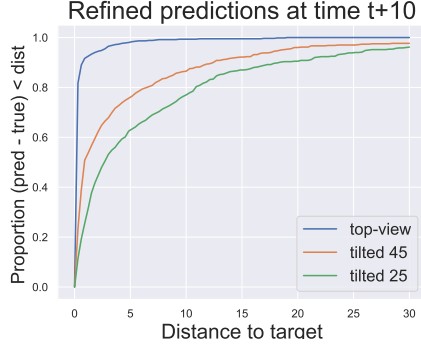

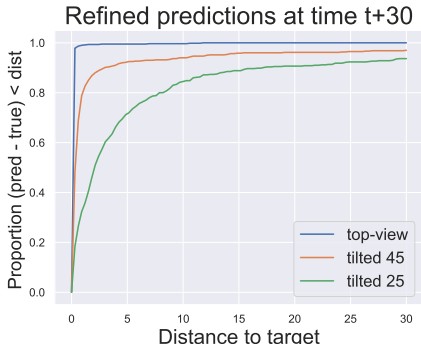

Figure S1. Proportion of correctly followed objects (y-axis) as a function of the distance error in pixels (x-axis) for our approach using rollout followed by the occlusion-aware position refinement. The different plots correspond to rollout lengths of 5 (left), 10 (middle) and 30 (right) frames. Different curves correspond to different camera view angles (top-view, tilted 45 degrees and tilted 25 degrees). In this experiment, all objects have the same shape and color making the task of following the same object for a long period of time very challenging. The plots demonstrate the success of our method in this very challenging scenario with object collisions and inter-object occlusions. For example, within a distance threshold of 20 pixels, which corresponds to the size of the smallest objects in the environment, our approach correctly follows more than 90% of objects during the rollout of 30 frames in all three considered camera viewpoints (top-view, 45 degrees and 25 degrees). **Please see also the supplementary videos "one_class*.mp4".**

R. Riochet, M. Ynocente Castro, M. Bernard, A. Lerer, R. Fergus, V. Izard, and E. Dupoux. IntPhys: A Framework and Benchmark for Visual Intuitive Physics Reasoning. *ArXiv e-prints*, March 2018.

Nicholas Watters, Andrea Tacchetti, Theophane Weber, Razvan Pascanu, Peter Battaglia, and Daniel Zoran. Visual Interaction Networks. *CoRR*, abs/1706.01433, 2017. URL http://arxiv.org/abs/1706.01433.

