# OpenReview forum: "Occlusion  resistant  learning  of  intuitive physics from videos"
_ICLR.cc/2020/Conference — Reject_

### Official Review · AnonReviewer2 · 2019-10-23
**Official Blind Review #2**

**Rating:** 3

**Review:**

Summary:
This paper proposes a method that combines a recurrent neural network that predicts values that are used as inputs to a rendered which interprets them and generates an object shape map and a depth map for every step of the dynamics predicted by the recurrent neural network. The proposed method is able to handle object occlusions and interactions. In experiments, the authors show improved performance against baselines for future prediction, object tracking, and object permanence.


Pros:
+ Rendering network used with RNN
+ Outperforms chosen baselines

Weaknesses / comments:
- Compositional Rendering Network has to be pretrained: Did the authors try to train the model end-to-end? It would be interesting to see if this can be done so the proposed network is more unified.

- Figure 3 is not self explanatory: It would be good if the authors add labels to the predicted and gt frames. It is not easy to parse this figure from just looking at it.

- Difference from Battaglia et al., 2016: It seems that the only difference between the proposed method and this baseline is the change of input/outputs (including output with variance), and training in full sequence (RNN)? This looks like a minor change to me and reduces the novelty of the proposed method.

- Table 1 (trained on ground-truth positions): The authors claim that their network performs similar to Battaglia et al., 2016, but it seems that the baseline is better than the proposed method for the short term predictions with a relative improvement of about 20% and for long term when the baseline is better (half of the tests)  it’s by a relative improvement of about 10%. Can the authors comment on this? Am I missing something?

- Implausibility score: What do the authors mean by “the maximum error through the whole sequence”? How is this defined?

- The authors compare with Riochet et al., 2018 in Table 4, but not in the rest of the evaluations. Can the authors comment on why this is the case?


Conclusion:
The paper proposes an interesting method, dataset, and seems to perform baselines in the quantitative evaluation. To the best of my knowledge, the current state of the method is novel in the rendering network. However, the rest of components have limited novelty. In addition, I have some comments about the paper which stated above.


**Experience Assessment:**

I have read many papers in this area.

**Review Assessment: Checking Correctness Of Derivations And Theory:**

N/A

**Review Assessment: Checking Correctness Of Experiments:**

I carefully checked the experiments.

**Review Assessment: Thoroughness In Paper Reading:**

I read the paper thoroughly.

---

> ### Author Response · Authors · 2019-11-15
> **Answer to Reviewer#2**
>
> End-to-end training: Yes we tried to train end-to-end, but generalization wasn’t as good, and it was also longer to train (by a factor of 10). The resulting masks and trajectories were similar but generalization to longer rollouts was qualitatively not convincing. It seems therefore that the renderer trains faster than the prediction network, and that an incorrect initial renderer is detrimental to prediction learning. This pleads for a kind of a curriculum learning (giving more weights initially to rendering accuracy), but this would require many more experiments that we thought would be better suited to a followup study.
>
> Remark on Figure 3: Thank you, we will update Figure 3 to make clearer and self-explanatory.
>
> Difference from Battaglia et al.: Training in full sequence (RNN) without the supervised velocity was not explored in previous work. Interaction Networks are being widely used in synthetic dataset where ground truth velocity and position is known. Showing that velocity can be kept as a latent variable, using only supervision of the a sequence of positions to train the Interaction Networks opens doors to real world applications where position is much easier to get than velocity.
>
> Claim on similar performance with Battaglia et al: We will soften this claim, saying that our approach performs similar to Battaglia et al. on long rollouts. Also note that on short rollouts, object dynamics are close to linear and the fact that the baseline is uses ground truth velocity may explain the gap in performance.
>
> Concerning the IntPhys benchmark, and the “maximum error through the whole sequence”. The mask prediction is done through the whole sequence. For each frame, we get the reconstruction loss. We give to each sequence a loss corresponding to the maximum loss of its frames (maximum across time).
>
> Concerning comparison with Riochet et al. on other tasks, it is difficult to compare on the state prediction (predicting location of each object). However it is possible to compare on mask prediction, which we did but kept in the supplementary material (see table S1). In this table we can see that our method performs better than baseline Riochet et al., trained on future mask prediction.

---

### Official Review · AnonReviewer5 · 2019-11-05
**Official Blind Review #5**

**Rating:** 3

**Review:**

This paper proposes a method to predict future trajectories by modeling partial and full occlusions. Although it is well-written and the topic sounds interesting, I failed to catch why this approach is required for this setting. So, to strengthen the message of this paper, I listed a couple of suggestions and comments below (from the most important to the least important):


1. It is a bit hard to catch how this model handles "diversity." Specifically, when predicting the futures, it should be able to generate stochastic outputs. However, I failed to find how diverse the output of the model is. If the output is not that stochastic, then it would be tough to believe that the model can "predict" the future; instead, it may "extrapolate" the current condition only. To reassure such concerns, I recommend reporting how diverse your output is. (One easy way is to report the variance of the predicted center mass values between multiple samples while reporting the l2 distance.)


2. For the future prediction task, it would be much better if it is compared with various state-of-the-art future prediction approaches [1, 2, 3, 4, 5, 6]. For some of the models, it could not be able to compare directly with this approach (e.g., lack of 'center of mass' information). However, it would be still okay once it is compared with other state-of-the-art results without feeding some 3D information (e.g., provide projected 2D video as an input). By doing so, I believe the readers can easily catch (1) why it is better to predict physical interaction in 3D space (instead of directly predicting from a 2D space), and (2) also why predicting occlusion is essential in this problem setting.


3. Minor comments:
(a) It is a bit hard to catch how the author computes the "aggregate pixel reconstruction error" in Table S1. I recommend adding an equation number there to make it clear.

(b) There are a couple of missing references: the last sentence on page 4, the first paragraph in Supplementary, the last sentence in Supplementary page 3, etc.

(c) \citep is often misused. Please replace some inappropriate \citep with \citet.

(d) Please check the format of the reference, as well; currently, it has various styles even for the same source/conference.




------------------------------------------------------------------

[Some comments based on the authors' rebuttal]


I thank the authors for their thorough comments and detailed explanations for each question. I carefully read the whole (not just my part), but it didn't change my mind; it would be much better if the claim comes with a more directly comparable result.

Some additional comments:
Q1-comment) I think the limitation of "learning to extrapolate"-style video prediction approach is partially presented in Reviewer #2's claim as well. Therefore, in this context, I recommend the author to show a better result to reassure the reader's concern.

Q2-comment) I at least strongly recommend to add more experiments with other baselines, rather than relying mainly on the original model of the dataset. Although the input condition of a model could be different, I at least do believe that it will help the readers to catch the benefit of your setting more clearly.

I hope this review phase would make your paper more powerful.




[1] Liang et al., Dual Motion GAN for Future-Flow Embedded Video Prediction, in ICCV, 2017
[2] Denton and Fergus, Stochastic Video Generation with a Learned Prior, in ICML, 2018
[3] Wichers et al., Hierarchical Long-term Video Prediction without Supervision, in ICML, 2018
[4] Wang et al., Video-to-Video Synthesis, in NeurIPS, 2018
[5] Heish et al., Learning to Decompose and Disentangle Representations for Video Prediction, in NeurIPS, 2018
[6] Minderer et al., Unsupervised Learning of Object Structure and Dynamics from Videos, in NeurIPS, 2019

**Experience Assessment:**

I have published in this field for several years.

**Review Assessment: Checking Correctness Of Derivations And Theory:**

I assessed the sensibility of the derivations and theory.

**Review Assessment: Checking Correctness Of Experiments:**

I assessed the sensibility of the experiments.

**Review Assessment: Thoroughness In Paper Reading:**

I read the paper at least twice and used my best judgement in assessing the paper.

---

> ### Author Response · Authors · 2019-11-15
> **Answer to Reviewer#5**
>
> Thank you for your helpful comments on the writing as well as the missing references which we will include.
>
> On stochasticity: Our model is not stochastic. The use of the word Proba-RIN may be misleading, but we only predicts a term of error (or uncertainty). One way to make it stochastic and have various possible outcomes would be to sample predictions from this distribution (centered around the prediction, with std equal to predicted uncertainty). We mainly used this term of error to stabilize learning and represent uncertainty, not to model possible alternative outcomes.
>
> Comparison with state of the art video prediction approaches:  Indeed it is interesting to compare our results with video prediction models. To that aim we compare with Riochet et al. baseline, trained to predict future video frames in a similar setup, and which predicts segmentation masks which we can compare with our approach. Other models that directly produce RGB images are difficult to compare because they predict in a different space where colour and texture rendering matters a lot. A nice thing with predicting in mask space is that it enables to concentrate on issues of position and interaction.

---

### Official Review · AnonReviewer3 · 2019-11-07
**Official Blind Review #3**

**Rating:** 3

**Review:**

The key contribution of this paper is a model that can predict the dynamics of pre-segmented image patches under multiple frames of occlusion. The input image is processed by a CNN, the dynamics are predicted by a recurrent interaction net, and the output image is generated by a (deconv) CNN.

The key weaknesses I see are:

- The objects must be pre-segmented by some externally defined mechanism. Where does this mechanism come from? Segmenting the objects is challenging, and there are various recent methods that explore how to learn to do this (van Steenkiste et al., 2018). But if one has the segmentation masks, that simplifies things considerably and also offers a good estimate of the location and velocity (if there are 2+ frames).

- During training, the error is computed on all frames, including occluded ones, and backpropagated into the weights. But if I understand this correctly, this means that for training you need access to ground truth rendered trajectories. It would be better if the model didn't require the ground truth segmentations for objects that are occluded. How would they be made available to a learning system?

- Generally the writing wasn't that clear and I struggled to understand some details of the model and training procedure.


Overall I don't believe this work is ready for publication, as there isn't that much novelty and the requirements are impractical.

**Experience Assessment:**

I have published in this field for several years.

**Review Assessment: Checking Correctness Of Derivations And Theory:**

I assessed the sensibility of the derivations and theory.

**Review Assessment: Checking Correctness Of Experiments:**

I assessed the sensibility of the experiments.

**Review Assessment: Thoroughness In Paper Reading:**

I read the paper at least twice and used my best judgement in assessing the paper.

---

> ### Author Response · Authors · 2019-11-15
> **Answer to Reviewer#3**
>
> Yes, we agree that our paper does not tackle prediction from images, but from segmentation masks. Here, we use ground truth masks, which is possible because of the synthetic nature of the dataset. Application to real videos would require to use a segmentation system. While it is true that initial position and velocity can be estimated from a pair of masks, the task we are tackling is still not trivial. First, during total or partial occlusions, position estimates are incorrect or missing. Second, masks correspondence across frames is NOT provided, which makes it challenging to recover full trajectories, especially across long occlusions. Both of these challenges are tackled by our use of the neural renderer.
>
> On the need of ground truth segmentation of occluded objects: The model does not have access to the segmentation masks of occluded objects (at any time). During training, the position of partially occluded object is refined thanks to the renderer. Fully occluded object have no corresponding mask and therefore the refinement does not modify anything (no gradient loss). We will clarify this in the rewrite.

---

### Official Review · AnonReviewer6 · 2019-11-08
**Official Blind Review #6**

**Rating:** 6

**Review:**

* Note: emergency review, done under a shorter time frame than good reviews require.

In this paper, the authors develop a highly structured model to predict motions of objects defined by segmentation masks and depths. The model trains a physics model (in the form of a slightly modified interaction network) and a renderer composed of a per-object renderer combined with an occlusion model which composes the per-object segmentation and depth into a scene segmentation and depth.

Positives:
- The jury is still out on the degree of structure required to do proper object processing (neural nets with large amounts of data, mildly structured nets like networks with attention, more structured nets like this, or a full fledged renderer-like probabilistic program); this work contributes novel work to the line of research which attempts to do object-level processing with structured models while still leveraging the power of neural networks.

- The experimental section appears very thorough and convincing, even if the dataset is relatively simple.

Negatives:
- The model requires highly privileged information (segmentation mask, depth) at training and test time. Given that the segmentation/depth data are not too far from the actual images, it would have been interesting to see if it were possible to work with pixels (a variant of the occlusion model would probably still work), at least at test time.

- Regarding using segmentation/depth as input to the model: for a real dataset, segmentation is more relevant: it is both less informative than positions (due to significant occlusions) and easier to measure. In this highly synthetic dataset, this feels more debatable: objects are more entangled in the segmentation (which makes using segmentation more challenging), but only weakly, with many frames with no occlusion; furthermore, segmentation provides object shapes as information.

- The paper is generally well written, but could benefit from some reorganization - instead of defining each module separately, it would be better to describe the flow of information through different modules, then describe the module. I was wondering for a while how the initial positions were estimated (required as input to both the interaction net and the renderer), but this only comes at the end of the paper.

- Some ablation experiments felt missing, for instance, the importance of the refinement network (also unfortunate that the details of refinement were not given in the main body).

- The stochasticity of the interaction network appears a bit weak (simple Gaussians) - it would be interesting to display some data to see if the ground truth data is indeed Gaussian like .

- Missing potential references:
Sequential Attend, Infer, Repeat: Generative Modelling of Moving Objects
Learning to Decompose and Disentangle Representations for Video Prediction
MONet: Unsupervised Scene Decomposition and Representation

**Experience Assessment:**

I have published one or two papers in this area.

**Review Assessment: Checking Correctness Of Derivations And Theory:**

I assessed the sensibility of the derivations and theory.

**Review Assessment: Checking Correctness Of Experiments:**

I assessed the sensibility of the experiments.

**Review Assessment: Thoroughness In Paper Reading:**

I made a quick assessment of this paper.

---

> ### Author Response · Authors · 2019-11-15
> **Answer to Reviewer#6**
>
> Thank you for your helpful comments on the writing as well as the missing references.
>
> We agree with your assessment that segmentation masks would be more difficult to obtain in real videos; probably, also the depth information would be less reliable than in the synthetic datasets we are using. Still, the main point of this paper is that a level of representation like object masks+depth, a kind of a 2.5D representation is a good level of representation to compute long term predictions, provided that occlusion can be addressed. Now, we agree that the next steps should address what happens when noisy masks are used instead of ground truth ones.
>
> Ablation studies: We agree that additional ablation studies on the refinement network are interesting. Following suggestions from Reviewer#6, we trained the same Recurrent Interaction network on a forward prediction task with 15 and 25 degrees tilted views, without using position refinement with the renderer. Hence the locations of objects used as input of the renderer exactly match the centroid of each segmentation mask. Note that we choose the 15 and 25 degrees tilted views so that occlusions occur frequently (otherwise refinement is not so useful). On the 25 degree tilted view, we obtain a L2 distance (in pixels) to target of 13.1/23.8 for a prediction horizon of 5 and 10 frames respectively. On the 15 tilted view, we obtain a L2 distance to target of 18.5/31.2 for prediction horizon of 5 and 10 frames. These results are to be compared with those in Table1 in the main body. We observe that in such an environment where occlusions appear frequently, refining estimates positions with the renderer reduces prediction errors by about 30%, which confirms intuition coming from qualitative results in supplementary material.
> Stochasticity: the stochasticity of the model is indeed restricted to the prediction of an uncertainty term. It seems hard to compare this with the stochasticity of the ground truth data since this data is (almost) deterministic. One way to do it would be to add white noise to the input data and estimate the distribution of the resulting trajectories. This indeed looks interesting. In the present model, we have observed that in long rollouts behind occlusions uncertainty increases, as well as after objects contact (bouncing events).

---

### Decision · Program_Chairs · 2019-12-19

**Decision:**

Reject

**Comment:**

The paper studies the problem of modeling inter-object dynamics with occlusions. It provides proof-of-concept demonstrations on toy 3d scenes that occlusions can be handled by structured representations using object-level segmentation masks and depth information. However, the technical novelty is not high and the requirement of such structured information seems impractical real-world applications which thus limits the significance of the proposed method.